# The Role of Membrane-Type 1 Matrix Metalloproteinase–Substrate Interactions in Pathogenesis

**DOI:** 10.3390/ijms24032183

**Published:** 2023-01-22

**Authors:** Hyun Lee, Lucas Ibrahimi, Dimitri T. Azar, Kyu-Yeon Han

**Affiliations:** 1Department of Pharmaceutical Sciences, College of Pharmacy, University of Illinois at Chicago, Chicago, IL 60612, USA; 2Biophysics Core at the Research Resources Center, University of Illinois at Chicago, Chicago, IL 60607, USA; 3Department of Ophthalmology and Visual Sciences, Illinois Eye and Ear Infirmary, College of Medicine, University of Illinois at Chicago, Chicago, IL 60612, USA

**Keywords:** MT1-MMP, MMPs substrates, proteolysis, VEGFR-1, FGFRs, angiogenesis

## Abstract

A protease is an enzyme with a proteolytic activity that facilitates the digestion of its substrates. Membrane-type I matrix metalloproteinase (MT1-MMP), a member of the broader matrix metalloproteinases (MMP) family, is involved in the regulation of diverse cellular activities. MT1-MMP is a very well-known enzyme as an activator of pro-MMP-2 and two collagenases, MMP-8 and MMP-13, all of which are essential for cell migration. As an anchored membrane enzyme, MT1-MMP has the ability to interact with a diverse group of molecules, including proteins that are not part of the extracellular matrix (ECM). Therefore, MT1-MMP can regulate various cellular activities not only by changing the extra-cellular environment but also by regulating cell signaling. The presence of both intracellular and extra-cellular portions of MT1-MMP can allow it to interact with proteins on both sides of the cell membrane. Here, we reviewed the MT1-MMP substrates involved in disease pathogenesis.

## 1. Introduction

Matrix metalloproteinases (MMPs) are a family of zinc-dependent endopeptidases, and there are currently 24 known MMPs in humans [1]. MMPs are expressed in most tissues during various cellular events, including tissue remodeling and wound healing [2,3]. These MMPs can be categorized into five groups based on their substrate specificity and localization, including (i) gelatinases (MMP-2 and -9), (ii) stromelysins (MMP-3, -10, and -11), (iii) collagenases (MMP-1, -8, and -13), (iv) matrilysins (MMP-7 and -26), and (v) membrane-type (MT)-MMPs (MMP-14, -15, -16, -17, -24, and -25) [4].

Membrane-anchored MMPs are more likely than most soluble MMPs to interact with a diverse group of molecules, including cellular receptors [5,6]. Membrane-anchored proteins have both intracellular and extracellular portions, which allow these proteins to interact with proteins on both sides of the membrane, whereas soluble MMP proteins are constrained to one side or the other. MMPs share similar structural characteristics, such as a signal sequence, propeptide, catalytic domain, hinge region, and hemopexin-like domain. Membrane-type matrix metalloproteinases (MT-MMPs) contain additional domains, such as the transmembrane domain and a cytoplasmic tail at the C-terminus (Figure 1A) [7,8]. MT-MMPs can be further categorized into type I (MT1-MMP, MT2-MMP, MT3-MMP, and MT5-MMP) and type II (MT4-MMP and MT6-MMP) [9]. MT1-MMP, also known as MMP-14, is important for the remodeling of the extracellular matrix during various cellular processes, including angiogenesis and wound healing, through its proteolytic activity. Zinc and calcium ions are important for maintaining the enzyme activity of MT1-MMP (Figure 1B). The interaction between a short cytoplasmic domain of MT1-MMP and MT1-MMP cytoplasmic tail binding protein 1 (MTCBP-1) leads MT1-MMP localization into the edge of the cell to clear a path during migration, affecting pancreatic tumor cell metastasis, in particular [10]. Therefore, MT1-MMP has more diverse functions than most MMPs due to its extra- and intracellular domains, which can interact with the extracellular matrix (ECM) and non-ECM molecules. This article’s purpose is to review various MT1-MMP substrates and determine how they relate to the disease’s development, such as arthritis, inflammation, and neovascularization (NV). 

## 2. Substrates of MT1-MMP

### 2.1. ECM Substrates

The ECM is an extensive molecular network composed of proteins, glycosaminoglycans, and glycoconjugates [11,12]. The central role of the ECM is to provide structural support to the surrounding cells. The most abundant ECM components are collagen and fibronectin, and their degradation by MT1-MMP results in enhanced cell migration during angiogenesis and inflammation [13,14,15,16,17,18]. Therefore, MMPs regulate the cell environment via the proteolytic degradation of ECM components.

#### 2.1.1. Collagen

Collagen is critical for maintaining the optimal structure of cells and tissues, including the skin, tendon, and bone. Collagen can be categorized into at least five types, type I to V, and collagen types I is mostly digested by MT1-MMP [19]. The glycine substitution mutation of collagen in the triple helical domain leads to congenital disorders such as osteogenesis imperfecta, which can also be caused by splicing mutations in *COL1A1/2*, disrupting the collagen functions [20,21]. In addition, excessive collagen degradation can not only increase the risk of developing pathophysiological conditions, such as arthritis and osteoarthritis [22] but can also increase metastasis in breast cancer [23]. Therefore, proper collagen homeostasis, synthesis, and degradation are important to promote healthy conditions. Most MMPs cannot cleave fibrillar collagen due to the tensile strength of the microfibrils through the combination of extensive cross-link and a right-handed supertwist structure [24]. Interestingly, MT1-MMP has a smaller catalytic pocket than other MMP collagenases, enabling the cleavage of glycine–leucine covalent bonds in collagen and leading to the denaturation of the rigid form of collagen into gelatin, which can be further cleaved by gelatinases, such as MMP-2 and MMP-9 [25]. A complex structure of the MT1-MMP hemopexin-like domain with a transient collagen triple helix has been solved by the solution of nuclear magnetic resonance (NMR) (Figure 2A) [26]. 

Furthermore, the expression of MT1-MMP in Madin-Darby canine kidney (MDCK) cells makes them rapidly invade across the type I collagen gels in comparison to the control MDCK cells. However, neither the cytoplasmic tail deleted form of MT1-MMP nor the transmembrane-deleted MT1-MMP-expressed MDCK enhanced the degradative potential of collagen [27]. Zigrino et al. reported that MT1-MMP–deficient mice showed highly accumulated collagen in soft tissues, indicating that MT1-MMP is a major collagenase that initiates collagen degradation [28]. Furthermore, MT1-MMP–knockout mice exhibited severe impairments in bone formation and vascular development and developed atherosclerotic plaques due to excessive collagen accumulation [29,30]. These results suggest that the proteolytic activity of MT1-MMP is important not only for collagen degradation but also for the development of various organs [19,29,31,32]. 

#### 2.1.2. Fibronectin

Fibronectin is a ~220 kDa glycoprotein and a component of the ECM. Fibronectin forms a complex with collagen to create a structural network to support cells and tissues. The primary role of fibronectin is to enhance the adhesion of neutrophils and endothelial cells, which is an essential step during wound healing [33,34]. In addition, the fibronectin arginine-glycine-aspartate (RGD) sequence interacts with cell surface molecules, such as integrin α5β1 and α4β1 (Figure 2B) [35], which are regulators of the actin assembly and upstream of small Rho GTPase–mediated signal transduction pathways [36,37,38]. Deletion of the fibronectin gene leads to early embryonic lethality due to a failure in heart development [39]. Fibronectin degradation and turnover are essential for cell migration during tumor metastasis, morphogenetic movement, and trophoblast implantation [40].

A robust fibronectin matrix was observed in cultured MT1-MMP-null myofibroblasts, whereas the overexpression of MT1-MMP in the same myofibroblasts led to the disruption of fibronectin accumulation [41]. In addition, several fibronectin fragments have been identified as being generated by MT1-MMP activity in synovial fluid from patients with arthritis [42]. These results support fibronectin as a substrate of MT1-MMP. Interestingly, a generated ~30 kDa fibronectin fragment by MT1-MMP acts as a pro-inflammatory factor, similar to interleukin (IL)-6 and IL-8 [43]. Therefore, MT1-MMP may facilitate the development of rheumatoid arthritis via the generation of pro-inflammatory fibronectin fragments that enhance the invasion of synovial fibroblasts into the cartilage [44].

#### 2.1.3. Aggrecan

Aggrecan is found within aggregates containing hyaluronan filaments and linked proteins, which provide joint lubrication in the articular cartilage [45,46]. A high concentration of aggrecan is essential for maintaining healthy articular cartilage. Aggrecan is susceptible to cleavage by various proteases, such as serine proteases, cysteine proteases, aspartic proteases, and MMPs. MMP-8 was the first MMP that was identified as being responsible for degrading aggrecan [47]. However, MT1-MMP has demonstrated higher cleavage efficiency than MMP-8 due to the three cleavage sites between the G1 and G2 domains, one of which has a unique recognition site (Figure 3) [48]. 

Aggrecan protects collagen from proteolytic proteases by developing a complex form [49]. Therefore, the cleavage of aggrecan is critical for initiating collagen degradation in the cartilage, in particular [50]. Elevated MT1-MMP expression was observed as a result of the expression of pro-inflammatory cytokines, such as IL-1 and tumor necrosis factor α (TNF-α), in degenerating the cartilage matrix [51]. In addition, high expression patterns of MMP-2 and MT1-MMP were observed in newly formed condylar cartilage during embryo development (until E18.0), indicating that MT1-MMP was actively involved in the remodeling of cartilage [52]. Furthermore, in comparative immunodetection analysis among catalytic domains of three MMPs (MMP-3, -8, and MT1-MMP), MT1-MMP exhibited stronger aggrecanase activity than MMP-8 and -3 [48]. Taken together, these results suggest that induced MT1-MMP serves as an aggrecanase of aggrecan/collagen complex remodeling in progressive joint disorders, such as osteoarthritis [53].

#### 2.1.4. Laminin

Laminins are non-collagenous glycoproteins that comprise three disulfide-linked polypeptide chains (α, β, and γ). Laminin chains are assembled by five α (α1–α5), three β (β1–β3), and three γ (γ1–γ3), resulting in more than 50 possible heterotrimeric αβγ assemblies in humans (Figure 4) [54,55]. The N-terminus of laminin forms a complex with other ECM molecules, thus playing a key role in the formation of basement membranes, which separates the epithelial tissue from the underlying connective tissue, whereas the C-terminus binds to cellular receptors that regulate cell growth and motility by activating signaling pathways; therefore, laminins serve as a link between intracellular signaling and the ECM [56,57,58]. 

The proteolytic modification of laminin chains alters the heterotrimer assembly, resulting in changes in cell migration. For example, plasmin enhances cell attachment to the matrix by cleaving the α3 subunit of laminin-5 (α3β3γ2) [59,60,61,62]. By contrast, MT1-MMP causes the cell to detach from the matrix by cleaving the laminin-332 and, therefore, enhancing cell migration (Figure 4) [63,64,65]. In a parallel experiment, epithelial cell migration was controlled by MT1-MMP via the cleavage of the laminin-511 [66]. Therefore, the MT1-MMP cleavage of laminins is an important proteolytic activity for migration, specifically at the leading edge of cells. Importantly, Gialeli et al. discovered that laminin’s excessive digestion by MT1-MMP can promote cancer progression [67]. Furthermore, MT1-MMP deficiency had smaller and heterogenous myofiber-like effects in patients with muscular dystrophy, which was caused by defects in the laminin-based cell adhesion system [68,69]. These results indicate that MT1-MMP is a regulator of laminin-based cell migration and is crucial for specific tissue development [70].

#### 2.1.5. Other ECM Substrates

Syndecan-1 is a transmembrane heparan sulfate proteoglycan and interacts with various extracellular molecules, including fibronectin, collagen, and growth factors [71]. Therefore, syndecan-1 participates in diverse cellular activities and is dependent on the characteristics of binding proteins [72]. Syndecan-1 cleavage by proteases, MMPs, and heparinase can disrupt the interactions between syndecan-1 and binding proteins, changing syndecan-1-mediated cellular activity [73]. MT1-MMP cleaves the peptide bond between Gly^245^–Leu^246^ within a syndecan-1 juxtamembrane region, generating a soluble syndecan-1 fragment in humans. The shedding of syndecan-1 was promoted in MT1-MMP co-expressed with syndecan-1 human embryonic kidney (HEK) 293T cells, whereas cell-anchored syndecan-1 was reduced [74]. Interestingly, syndecan-1 accumulated on the cell surface in the presence of MMP inhibitors, batimastat (BB-94) or TIMP-2, in cells that were transfected with syndecan-1 cDNA, decreasing cell motility as a result [74]. Thus, MT1-MMP is a key enzyme for the shedding of syndecan-1 and regulates syndecan-1-mediated cell migration [75]. More importantly, a recent study showed that MT1-MMP is involved in promoting the proliferation of Basal-Like breast cancer cells through the cleavage of syndecan-1 [76]. 

Lumican is the primary keratan sulfate proteoglycan that is present in the corneal stroma and cartilage [77,78]. Lumican binds to collagen and maintains the spacing between adjacent collagen fibrils, which is critical for the proper alignment of collagens necessary for corneal transparency [79]. In an in vitro proteolysis assay, 85 kDa lumican disappeared after incubation with the catalytic domain of MT1-MMP but retained its size in the presence of an MMP inhibitor, BB-94. In addition, overexpressed lumican in HEK 293 cells increased the expression of tumor suppressor genes such as p21/Waf-1. This expression was reduced by the co-expression of MT1-MMP and lumican in the same HEK 293 cells [80]. Interestingly, lumican was also identified as a competitive inhibitor of the MT1-MMP [81]. Hence, lumican is a substrate of MT1-MMP and can also play a role as an inhibitor of MT1-MMP, and lumican-mediated activity can be regulated by MT1-MMP [82]. 

### 2.2. Non-ECM Substrates

#### 2.2.1. pro-MMP-2 and pro-MMP-13

Most MMPs consist of structurally conserved domains, one of which is the pro-domain with approximately 80 residues [83]. The C-terminus of the pro-domain interacts with zinc ions in the catalytic domain to restrain its catalytic activity [84,85,86]. Consequently, the intercalation of a water molecule into the catalytic site is prevented until the pro-domain is cleaved by furin or other proteases [87]. MMP-2 is involved in many cellular activities, such as angiogenesis, tissue repair, and inflammation, via the degradation of ECM molecules, some of which are collagen type IV, gelatin, elastin, and fibronectin [88]. Hence, impaired tumor-induced angiogenesis was observed in MMP-2-deficient mice, resulting in dramatically smaller tumor volumes compared to those in wild-type mice following identical tumor injections [89]. 

MT1-MMP has been reported to activate pro-MMP-2 at a faster rate compared to other proteases, such as MMP-1 and MMP-7 [90]. The formation of a ternary complex containing MT1-MMP, pro-MMP-2, and the tissue inhibitor of metalloproteinase 2 (TIMP-2) is required for the complete activation of pro-MMP-2 [91]. This ternary complex structure is not available to date, but the X-ray crystal structure of the catalytic domain of MT1-MMP complexed with TIMP-2 was reported a while ago (Figure 5) [92]. The MT1-MMP/TIMP-2 complex acts similarly to a receptor that binds with pro-MMP-2, facilitating the cleavage of the pro-domain of pro-MMP-2 by a TIMP-2–free neighboring MT1-MMP [93]. The processing of MMP-2 activation did not occur in TIMP-2 knockout cells, although some intermediate forms of MMP-2 were observed [94]. Decreased levels of activated MMP-2 as a result of the loss of MT1-MMP in vitro and in vivo have indicated that pro-MMP-2 is a substrate of MT1-MMP and has demonstrated the essentiality of the MT1-MMP/TIMP-2 complex to activate pro-MMP-2 [95]. On the other hand, MT1-MMP directly binds to pro-MMP-13 and is able to cleave between the Glu^84^–Tyr^85^ peptide bond in pro-MMP-13 without needing another protein, such as TIMP-2 [96]. The level of active MMP-13 decreased in aortic extracts receiving MT1-MMP^−/−^ bone marrow compared to those in the MT1-MMP^+/+^ bone marrow [97]. Type II collagen is processed by MMP-13, resulting in cartilage degradation [98]. Therefore, MT1-MMP can influence the MMP-13-mediated collagen degradation, leading to degenerative diseases such as osteoarthritis [99,100].

#### 2.2.2. Vascular Endothelial Growth Factor Receptor-1 and Fibroblast Growth Factor Receptor-2

The vascular endothelial growth factor (VEGF) is the most potent growth factor expressed in vascular endothelial cells and non-endothelial cells, including cancer cells [101]. VEGF has been identified as a vascular permeability factor; however, VEGF is best known as a potential mitogen that is involved in vascular development, bone formation, and wound healing through binding to VEGF receptors (VEGFRs) [102,103]. VEGFR-2 is a functional receptor, whereas VEGFR-1 is a non-functional receptor due to weak tyrosine kinase activity, despite a higher binding affinity for VEGFA compared to VEGFR-2. Therefore, VEGFR-1 acts as a decoy receptor that impedes VEGFA–VEGFR-2–mediated downstream signaling, maintaining the balance between VEGFR-1 and VEGFR-2 activity [104]. It is also known that VEGFR-1 is a receptor of other members of the VEGF family, such as VEGF-B and PIGF. We have reported that MT1-MMP only cleaves VEGFR-1, which may shift the balance toward VEGFR-2 activity. Consequently, the cleavage of membrane-anchored VEGFR-1 by MT1-MMP results in enhanced VEGFA–VEGFR-2–mediated extracellular signal-regulated kinase activation in endothelial cells [105,106]. Furthermore, elevated MT1-MMP levels have been reported in neovascular and wounded tissues. These results support the pro-angiogenic activity of MT1-MMP via the regulation of membrane-anchored VEGFR-1 levels (Figure 6A). 

Fibroblast growth factor 2 (FGF-2, also known as the basic fibroblast growth factor) is another essential growth factor involved in vascular development. FGF-2 is a member of the FGF family, which consists of 23 structurally related peptides and is a known pro-angiogenic factor [107,108]. Interestingly, FGF-2–induced corneal neovascularization is impaired in MT1-MMP–knockout mice [95]. In addition, FGF-2-induced proliferation was significantly lower in MT1-MMP^−/−^ osteoblasts than in normal cells, supporting defective parietal bone growth as a result of the defective response of osteoblasts regarding FGF-2 signaling in MT1-MMP–knockout mice. In a subsequent experiment, MT1-MMP indirectly influenced FGFR-2 expression levels via the proteolysis of a disintegrin and metalloprotease domain-containing protein 9 (ADAM-9), and ADAM-9 degraded membrane FGFR-2 (Figure 6B) [109]. MT1-MMP inactivated ADAM-9 through the shedding of the ectodomain. Taken together, these results suggest that MT1-MMP regulates FGF-2 signaling through ADAM-9.

### 2.3. Other Non-ECM Substrates

Apolipoproteins interact with lipids to form lipoproteins. Apolipoprotein A-1 (Apo A-1) is a major component of high-density lipoprotein (HDL) and plays an important role in reverse cholesterol transport, delivering excess cholesterol back to the liver [110,111]. Therefore, Apo A-1 protects the cardiovascular system and reduces cardiovascular disease risk by preventing the accumulation of cholesterol in vascular tissues [112]. The absence of Apo A-1 resulted in the reduction of HDL in plasma, raising the risk of coronary artery disease [113]. Apo A-1 as a substrate of MT1-MMP was supported by the presence of Apo A-1 fragments in plasma samples following their incubation with MT1-MMP. Apo A-1 fragments were compared with harvested plasma in the presence or absence of MT1-MMP using 2-D gel analysis. Newly observed protein bands following incubation with MT1-MMP were characterized as 27-, 22-, and 14-kDa Apo A-1 fragments by matrix-assisted laser desorption/ionization time-of-flight mass spectrometry [114,115]. These results opened up the possibility of a new role of MT1-MMP in atherosclerosis development [116].

CD44 is a multifunctional adhesion molecule involved in cell–cell or cell–matrix interactions [117]. Terawaki et al. solved a complex structure of the cytoplasmic tail domain of MT1-MMP with Radixin: part of the ezrin/radixin/moesin (ERM) proteins, demonstrating that MT1-MMP is a crucial enzyme in shedding CD44 involved in tumor cell invasion (Figure 7) [118]. An elevated level of soluble CD44 has been observed in the synovial fluid in arthritis compared to healthy fluid. This coincidentally increased the infiltration of immune cells into synovial tissue by breaking CD44-mediated cell adhesion [119,120]. MT1-MMP cleaves the stem region of CD44 (Thr^130^–Gln^265^), which is located between the globular and transmembrane domains, to promote the detachment of cells from the matrix [121]. Hence, silencing CD44 in Transformed African Green Monkey Kidney Fibroblast (COS-1) cells interfered with cell migration [122]. Interestingly, EGFR-mediated signaling was not activated in the silencing of CD44 in the same cells. These results indicate that MT1-MMP regulates CD44-mediated cell migration and disrupts the crosstalk between CD44 and EGFR, which is required for EGFR activation.

The receptor activator of nuclear factor kappa B (NF-κB) ligand (RANKL) is a homotrimeric transmembrane complex that is involved in the development of osteoclasts [123]. In order to activate RANK, a receptor for RANKL on the preosteoclasts, membrane-anchored RANKL has to be cleaved by proteolytic enzymes [124]. MT1-MMP is responsible for releasing a soluble active form of RANKL (sRANKL) and regulating RANK-mediated osteoclast differentiation and bone resorption [125]. Furthermore, levels of sRANKL were markedly reduced in MT1-MMP–deficient mice in comparison to wild-type mice, clearly supporting the role of MT1-MMP in bone development via the RANKL/RANK system [126]. In addition, MT1-MMP overexpressed prostate cancer cells secreted more sRANKL. A high number of prostate cancer patients exhibited skeletal metastasis associated with elevated expression levels of MT1-MMP and sRANKL [127,128]. Therefore, MT1-MMP acts as a RANKL sheddase, participating in bone differentiation via the RANK/RANKL system.

Heparin-binding epidermal growth factor (EGF)-like growth factor (HB-EGF) belongs to the EGF family of molecules [129]. Pro-HB-EGF (Pro-domain) is required to cleave the pro-domain by MT1-MMP to become an active growth factor [130,131]. In this process, pro-HB-EGF splits into two parts, one of which is a soluble HB-EGF (sHB-EGF) which is responsible for synergistically activating the EGFR pathway. On the other hand, a C-terminal HB-EGF translocates into the nucleus and activates gene expression, resulting in cell proliferation [132]. MT1-MMP cleaved the N-terminal of HB-EGF, generating the heparin-independent growth factor [133]. Lung cancer is associated with increased expression of MT1-MMP and HB-EGF [134]. MT1-MMP co-expressed with HB-EGF in ovarian cancer cells exhibited enhanced cell growth and spread in vivo [133,134]. These results indicate that MT1-MMP is actively involved in the EGFR pathway and EGF-mediated tumorigenesis. 

## 3. Summary

The importance of MT1-MMP has been supported by numerous studies investigating genetic mutations in MT1-MMP, both in vitro and in vivo, particularly due to the lethality observed in MT1-MMP–knockout mice after birth, associated with severe defects, including impairments in bone formation and vascular formation. The breakdown of collagen is a major function of MT1-MMP, and the loss of collagen turnover may result in immature bone development, explaining the smaller size of MT1-MMP–knockout mice relative to wild-type mice. In addition, high levels of collagen and fibronectin were observed in the cartilage and soft tissues in the in vivo models with MT1-MMP deficiency. These results demonstrate that the MT1-MMP regulation of ECM molecules is crucial for the remodeling of the ECM during the development of various organs.

Another interesting abnormality observed in MT1-MMP–knockout mice is immature vascular development, such as defective vascular invasion and an impaired FGF-2–mediated angiogenic response. MT1-MMP also plays an important role in angiogenesis. This is the process through which capillary vessels are generated from existing blood vessels to increase the blood supply. This helps meet the metabolic demands of expanding tissues. In this process, MT1-MMP enhances angiogenic responses not only by clearing the path for migrating cells but also by degrading non-ECM components, which increase the number of active forms for proangiogenic factors, such as ECM-tethered cytokines, one of which is a latent transforming growth factor-β (TGF-β) and their receptor-mediated angiogenesis. Upregulated MT1-MMP, with other proangiogenic factors such as MMP-2 and MMP-13, are associated with various progressive diseases, including tumor invasion, corneal neovascularization, and proliferative diabetic retinopathy via active neovascularization. Hence, identifying new MT1-MMP substrates will expand our understanding of the pathophysiological roles of MT1-MMP. Over the past decade, advanced omics-based techniques, proteomics, and metabolomics have been utilized to identify differences in the protein expression profiles between normal and MT1-MMP–deficient cells. An alternative approach is the use of MT1-MMP featuring a Glu240 to Ala point mutation (E240A), rendering it catalytically inactive. This mutant can be used to capture MT1-MMP binding molecules from various types of samples, including cell lysates and cDNA libraries from Hela cells. Finding novel MT1-MMP substrates will expand our knowledge and shed further light on the search for more critical roles of MT1-MMP under both normal and pathological conditions.

## Figures and Tables

**Figure 1 ijms-24-02183-f001:**
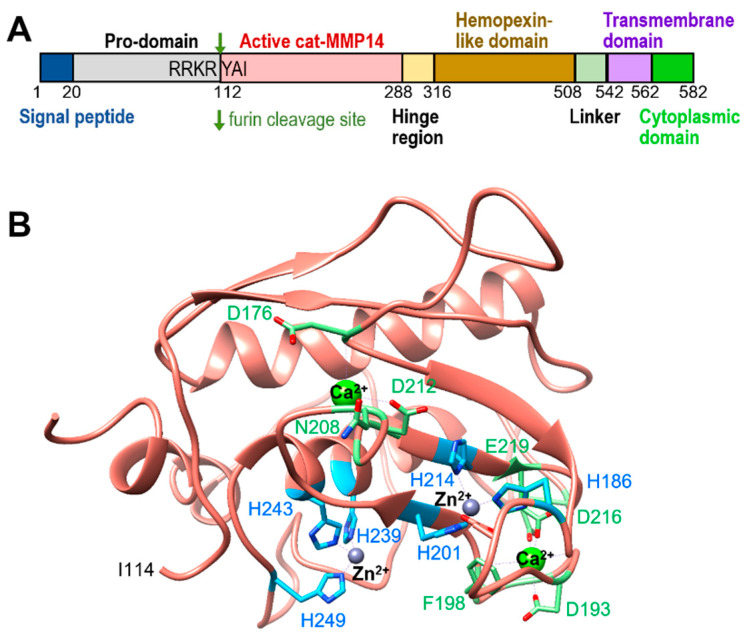
MT1-MMP background information. (**A**) A schematic representation of the full-length MT1-MMP. A furin cleavage site between pro-domain and cat-MT1-MMP is shown in dark green, and an active form of catalytic MMP-14 (residues 112–288) is in salmon color. The transmembrane and cytoplasmic domains are shown in purple and light green, respectively. (**B**) Structure of the MT1-MMP catalytic domain (PDB: 1BQQ). Two zinc ions (grey circle) and two calcium ions (lime green circle) are shown with residues that interact with them in blue and green color, respectively.

**Figure 2 ijms-24-02183-f002:**
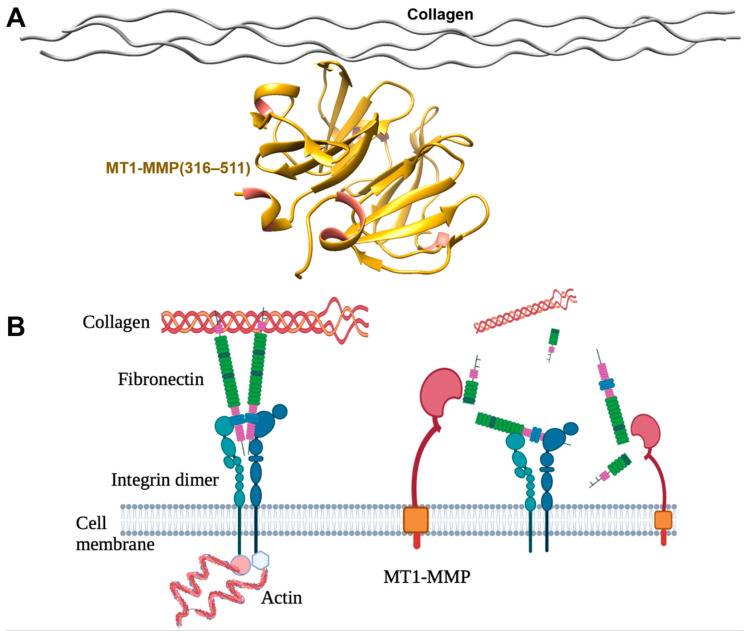
Collagen and fibronectin. (**A**) Solution nuclear magnetic resonance (NMR) structure (PDB code: 2MQS) of hemopexin-like domain of MT1-MMP (316–511) complexed with transient collagen triple helix. (**B**) Fibronectin complex with collagen and integrin during ECM remodeling. MT1-MMP cleaved fibronectin resulted in the disruption of interaction between collagen and integrin on the cell surface.

**Figure 3 ijms-24-02183-f003:**
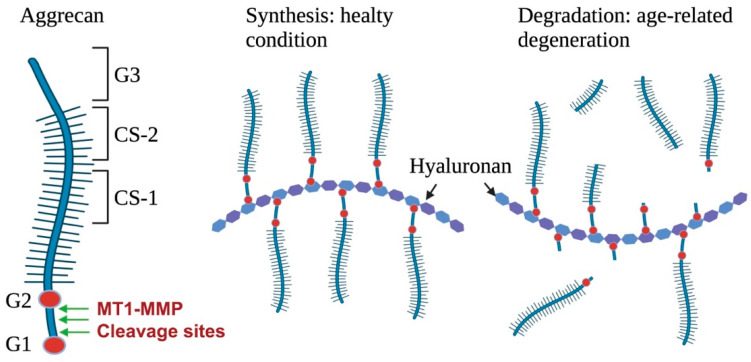
Structure of aggrecan and degradation. Aggrecan is a major proteoglycan in cartilage and plays an important role by forming a water-swollen matrix. MT1-MMP can cleave three positions between G1 and G2, which are essential binding motifs with the hyaluronan and link protein. MT1-MMP acts as an aggrecanase under pathophysiological conditions leading to the dehydration of articular cartilage.

**Figure 4 ijms-24-02183-f004:**
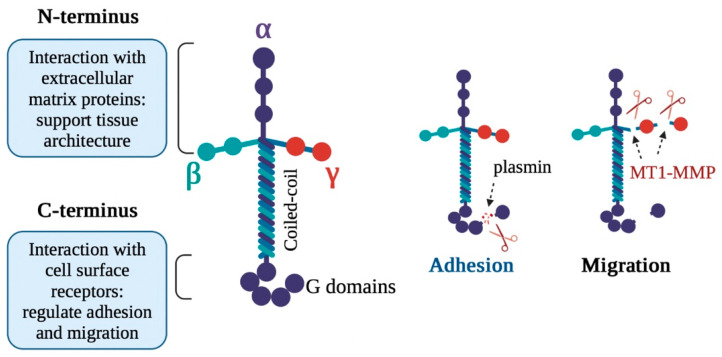
Structure of laminin and its role in cell migration. The N-terminus of laminin stabilizes the structure of cells and tissues following interaction with ECM molecules. The C-terminus of laminin can engage in cell movements. Cleavage of the α subunit leads to the enhanced attachment of cells to the matrix, whereas proteolysis of the γ subunit by MT1-MMP promotes the detachment of cells from the matrix, triggering migration.

**Figure 5 ijms-24-02183-f005:**
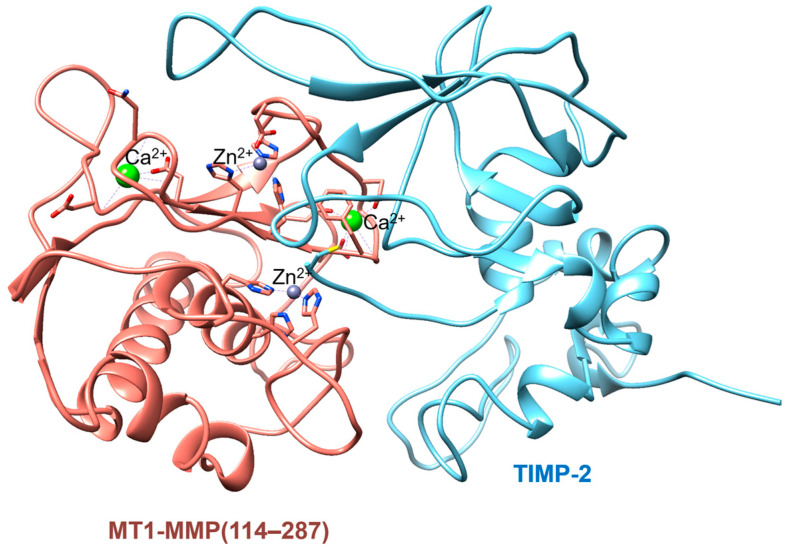
X-ray crystal structure of the catalytic domain of MT1-MMP (114–287) complexed with tissue inhibitor of metalloproteinase 2 (TIMP-2). The PDB code is 1BUV.

**Figure 6 ijms-24-02183-f006:**
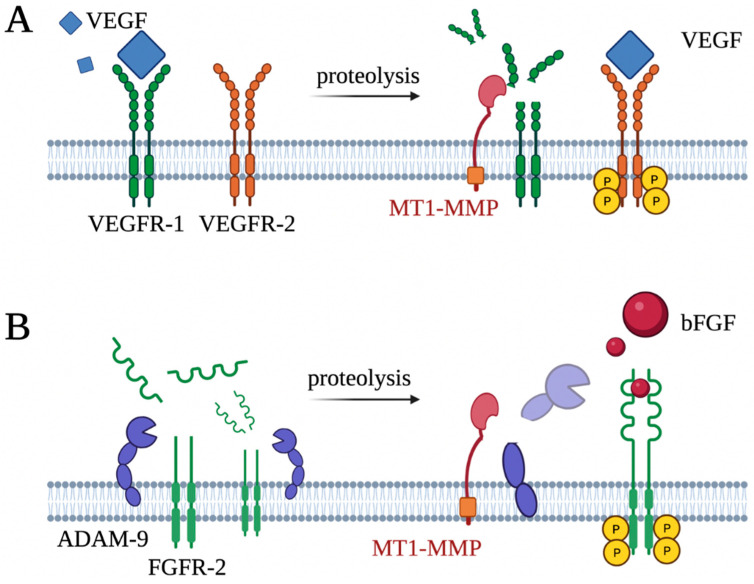
Potential pro-angiogenic role of MT1-MMP through regulation of VEGFR-2 and FGFR-2. (**A**) MT1-MMP is involved in VEGFA/VEGFR-2-mediated angiogenesis. VEGFR-1 exhibits a higher binding affinity with its ligand, VEGFA, than functional VEGFR-2. Membrane anchored MT1-MMP cleaved VEGFR-1, resulting in the enhancement of the interaction between VEGFA and VEGFR-2. (**B**) MT1-MMP regulates bFGF/FGFR-2-mediated angiogenesis. FGFR-2 is proteolyzed by ADAM-9, resulting in the elimination of the FGFR-2-downstream signal. MT1-MMP protects FGFR-2 by shedding an active ADAM-9 and restoring bFGF/FGFR-2-mediated angiogenesis.

**Figure 7 ijms-24-02183-f007:**
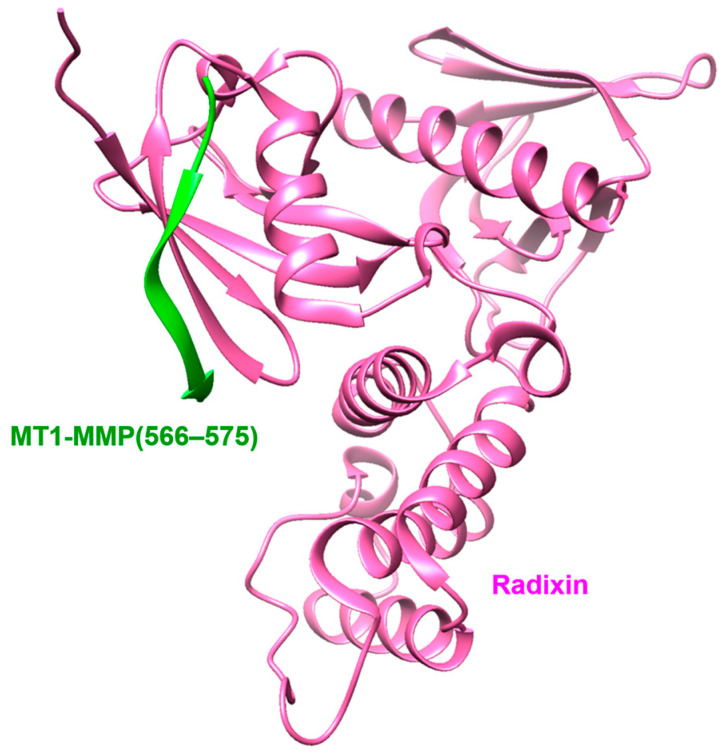
X-ray structure of the cytoplasmic domain of MT1-MMP (green) complexed with Radixin (pink): part of the ezrin/radixin/moesin (ERM) proteins. PDB code: 3 × 23.

## Data Availability

Not applicable.

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
