# Peer review of "The Role of Membrane-Type 1 Matrix Metalloproteinase–Substrate Interactions in Pathogenesis"

_ijms, 2023, doi:10.3390/ijms24032183_

Round 1

Reviewer 1 Report

This is a valuable review article in this field. I have just some specific comments as:
1. In Introduction chapter, authors should mention that MT1-MMP also known as MMP-14.

2. As there are several collagenases with specific type as collage as substrate, authors should mention at ,2.1.1 section, which type of collagen mostly digest by MT1-MMP.

3. Page 3, line 83, what is the MDCK cells? 
4. Authors did not include any example study about excessive collagen degradation by TM1-MMP and pathogenesis of that in cancer and inflammation.
5. In chapter 2.1.4. there is no example study about laminin excessive digestion by MT1-MMP which can promote cancer. For Instance, degradation of laminin-5 by MT1-MMP can generate cryptic peptides which can promote the cancer cells migration. 
Ref: Gialeli C, Theocharis AD, Karamanos NK. Roles of matrix metalloproteinases in cancer progression and their pharmacological targeting. The FEBS journal. 2011 Jan;278(1):16-27.

6. At chapter 2.1.5 author should mention about any study which indicate about disorders which result from Syndecan-1 cleavage by MT1-MMP. For example, they miss recently published article about this topic:

Ref: Nadanaka S, Bai Y, Kitagawa H. Cleavage of Syndecan-1 Promotes the Proliferation of the Basal-Like Breast Cancer Cell Line BT-549 Via Akt SUMOylation. Frontiers in cell and developmental biology. 2021:1210.

7. Some study showed that Lumican can behaves as a competitive inhibitor for MT1-MMP and block its activity. Authors should discuss in this matter.

Ref: Pietraszek K, Chatron-Colliet A, Brézillon S, Perreau C, Jakubiak-Augustyn A, Krotkiewski H, Maquart FX, Wegrowski Y. Lumican: a new inhibitor of matrix metalloproteinase-14 activity. FEBS letters. 2014 Nov 28;588(23):4319-24.

Author Response

RE: ijms-2165598

Title: The role of membrane-type 1 matrix metalloproteinase–substrate interactions in pathogenesis

Reviewer 1

This is a valuable review article in this field. I have just some specific comments as:
1. In Introduction chapter, authors should mention that MT1-MMP also known as MMP-14.

Author’s response: Thanks for pointing this out! It has been added to line 44 in introduction.

  1. As there are several collagenases with specific type as collage as substrate, authors should mention at ,2.1.1 section, which type of collagen mostly digest by MT1-MMP.

Author’s response: The following sentence has been added to section 2.1.1.

“Collagen can be categorized to at least five types, type I to V, and collagen type I is mostly digested by MT1-MMP [18].”

  1. Page 3, line 83, what is the MDCK cells? 

Author’s response: Madin-Darby canine kidney (MDCK) has been added to line 89. Thanks!

  1. Authors did not include any example study about excessive collagen degradation by TM1-MMP and pathogenesis of that in cancer and inflammation.

Author’s response: We now added an example case of breast cancer metastasis regulated by excessive collagen degradation and its corresponding reference in section 2.1.1.

  1. In chapter 2.1.4. there is no example study about laminin excessive digestion by MT1-MMP which can promote cancer. For Instance, degradation of laminin-5 by MT1-MMP can generate cryptic peptides which can promote the cancer cells migration. 

Ref: Gialeli C, Theocharis AD, Karamanos NK. Roles of matrix metalloproteinases in cancer progression and their pharmacological targeting. The FEBS journal. 2011 Jan;278(1):16-27.

Author’s response: Below sentence was added to section 2.1.4. Thank you so much for the reference! It is very helpful.

“Importantly, Gialeli et al. discovered that laminin excessive digestion by MT1-MMP can promote cancer progression [67].”

  1. At chapter 2.1.5 author should mention about any study which indicate about disorders which result from Syndecan-1 cleavage by MT1-MMP. For example, they miss recently published article about this topic:

Ref: Nadanaka S, Bai Y, Kitagawa H. Cleavage of Syndecan-1 Promotes the Proliferation of the Basal-Like Breast Cancer Cell Line BT-549 Via Akt SUMOylation. Frontiers in cell and developmental biology. 2021:1210.

Author’s response: The following sentence was added to section 2.1.5. “More importantly, a recent study showed that MT1-MMP is involved in promoting the proliferation of the Basal-Like breast cancer cells by cleavage of syndecan-1 [76].”

  1. Some study showed that Lumican can behaves as a competitive inhibitor for MT1-MMP and block its activity. Authors should discuss in this matter.

Ref: Pietraszek K, Chatron-Colliet A, Brézillon S, Perreau C, Jakubiak-Augustyn A, Krotkiewski H, Maquart FX, Wegrowski Y. Lumican: a new inhibitor of matrix metalloproteinase-14 activity. FEBS letters. 2014 Nov 28;588(23):4319-24.

Author’s response:  We appreciate the reviewer for telling us this interesting fact. We added the following sentence to section 2.1.5. “Interestingly, lumican was also identified as a competitive inhibitor of the MT1-MMP].”

Reviewer 2 Report

In their review article “The role of membrane-type 1 matrix metalloproteinase–substrate interactions in pathogenesis” the authors examine the role of MT1-MMP and various of its substrates during matrix remodeling in the pathogenesis of various diseases, ranging from arthritis and inflammation to cancer.

The authors briefly describe the structure of MT1-MMP and then discuss the proteolytic processing of its major substrates in the context of a broad spectrum of diseases. The authors focus on the ECM components collagen, fibronectin, aggrecan, syndecan and lumican, as well as other MMPs such as the proMMPs -2 and -13 and important signaling molecules such as VEGFR-1, FGF-2.

The review article is well designed and the manuscript is well-written, providing data supporting their conclusions. There are just some minor points to comment on.

Major comments and suggestions:

none

Minor points:

Line 17: MMPs -1, -8, and -13 are all collagenases. To my knowledge MMPs -8 and -13, but not the interstitial collagenase MMP-1 can be activated by MT1-MMP. Please specify, which one is meant here.

Line 28: Since there are two isoforms of MMP23 encoded at different loci, there are actually 24 MMPs in humans.

Line 40: Depending on their membrane anchoring, MMPs can be type I or type II transmembrane proteins. MT1-MMP is of type I. However, the wording here is not clear. Especially because of the abbreviation introduced in line 15.

Line 53: The signal sequence aa 1-20 is depicted in dark blue. To my knowledge, the propeptide ranges from aa 21-111 and contains a furin cleavage site at position 89. This is drawn but not labeled. What is the purpose of the light blue marking (aa 20-89)?

Line 70 ff: The most common pathogenic causes underlying osteogenesis imperfecta are  substitutions of another amino acid for glycine in the triple helical domain of either chain, whereas splice variants come second.

Line 79: Collagen-like gelatin? Maybe delete “collagen-like” here?

Line 82 and Fig. 2A: To my understanding, MT1-MMP folds over the collagen triple-helix to form a sandwich with the HPX domain and to embrace a collagen monomer between the domains for cleavage.

Line 83, 183, 193, 293, 353: MDCK, HEK293T, HEK  293, COS-1, Hela cells: please specify the full name/source of these cell lines.

Line 152: maybe replace “to influence the tissue architecture” by “,thus playing a key role  in the formation of basement membranes, which separate epithelial tissue from the underlying connective tissue”?

Line 159, 161: maybe use the names laminin-332 for laminin-5 and laminin-511 for laminin-10?

Line 185: Maybe mention the name batimastat for BB-94? Also note that this is not specific to MT1-MMP but a broad spectrum MMP inhibitor.

Line 197: Furthermore, lumican is a competitive inhibitor of collagen cleavage by MT1-MMP (PMID: 25304424).

Line 206: There are various collagen types. Not all of them are known to be substrates of MMP-2. Also, MMP-2 can cleave only monomeric collagen chains after at least partial unwinding of its triple helix.

Line 239: As described here, VEGFR-1 acts as a decoy receptor for VEGF-A; but it is also the exclusive receptor or other members of the VEGF family, such as VEGF-B and PlGF.

Line 262, Fig. 6: Why are small debris fragments shown only for ADAM-9 cleavage but not for VGFR-1 cleavage, and why are they red/blue colored? Also, in Figures A and B, why is the phosphorylated receptor shown again on a smaller scale at the far right?

Line 320: Which types of cancer are meant here? Also, a reference would be desirable.

Line 343: Please specify what is meant with “active forms of proangiogenic factors”: ECM-tethered cytokines released by matrix degradation and/or ECM fragments acting as matrikines?

Line 602: This citation is incomplete.

English is fine. There are only a few minor grammar mistakes, e.g., lines 222, 317 f.

Author Response

RE: ijms-2165598

Title: The role of membrane-type 1 matrix metalloproteinase–substrate interactions in pathogenesis

Reviewer 2

In their review article “The role of membrane-type 1 matrix metalloproteinase–substrate interactions in pathogenesis” the authors examine the role of MT1-MMP and various of its substrates during matrix remodeling in the pathogenesis of various diseases, ranging from arthritis and inflammation to cancer.

The authors briefly describe the structure of MT1-MMP and then discuss the proteolytic processing of its major substrates in the context of a broad spectrum of diseases. The authors focus on the ECM components collagen, fibronectin, aggrecan, syndecan and lumican, as well as other MMPs such as the proMMPs -2 and -13 and important signaling molecules such as VEGFR-1, FGF-2.

The review article is well designed and the manuscript is well-written, providing data supporting their conclusions. There are just some minor points to comment on.

Major comments and suggestions:

none

Minor points:

Line 17: MMPs -1, -8, and -13 are all collagenases. To my knowledge MMPs -8 and -13, but not the interstitial collagenase MMP-1 can be activated by MT1-MMP. Please specify, which one is meant here.

Author’s response: It has been corrected. We appreciate the reviewer.

Line 28: Since there are two isoforms of MMP23 encoded at different loci, there are actually 24 MMPs in humans.

Author’s response: Thanks for pointing this out! It has been corrected to 24.

Line 40: Depending on their membrane anchoring, MMPs can be type I or type II transmembrane proteins. MT1-MMP is of type I. However, the wording here is not clear. Especially because of the abbreviation introduced in line 15.

Author’s response: We agree with the reviewer that it is unclear. It has been revised to the following sentence. “Membrane type matrix metalloproteinases (MT-MMPs) contain additional domains like transmembrane domain and a cytoplasmic tail at the C-terminus (Figure 1A) [7, 8]. MT-MMPs can be further categorized into type I (MT1-MMP, MT2-MMP, MT3-MMP, and MT5-MMP) and type II (MT4-MMP and MT6-MMP) [9].”

Line 53: The signal sequence aa 1-20 is depicted in dark blue. To my knowledge, the propeptide ranges from aa 21-111 and contains a furin cleavage site at position 89. This is drawn but not labeled. What is the purpose of the light blue marking (aa 20-89)?

Author’s response: We removed the linker region between signal peptide and pro-domain for we agree that it might be unnecessarily confusing, but we believe furin cleavage site is between pro-domain and catalytic domain, releasing the catalytic domain free from the pro-domain. Please correct us if we are wrong. We can revise it accordingly. We thought that furin recognizes 108RX(K/R)R111 and cleavage location is between two residues, R111 and Y112.

Line 70 ff: The most common pathogenic causes underlying osteogenesis imperfecta are  substitutions of another amino acid for glycine in the triple helical domain of either chain, whereas splice variants come second.

Author’s response: We appreciate the reviewer for pointing this out. We revised it to the following sentence. “The glycine substitution mutation of collagen in the triple helical domain leads to con-genital disorders like osteogenesis imperfect, which can also be caused by splicing mutations in COL1A1/2, disrupting the collagen functions”

Line 79: Collagen-like gelatin? Maybe delete “collagen-like” here?

Author’s response: Deleted as suggested.

Line 82 and Fig. 2A: To my understanding, MT1-MMP folds over the collagen triple-helix to form a sandwich with the HPX domain and to embrace a collagen monomer between the domains for cleavage.

Author’s response: The figure 2A is the only deposited structure of the hemopexin-like domain of MT1-MMP (316-511) complexed with transient collagen triple helix in Protein Data Bank (PDB code: 2MQS).

Line 83, 183, 193, 293, 353: MDCK, HEK293T, HEK  293, COS-1, Hela cells: please specify the full name/source of these cell lines.

Author’s response: Full names have been added to appropriate places. Thanks!

Line 152: maybe replace “to influence the tissue architecture” by “,thus playing a key role  in the formation of basement membranes, which separate epithelial tissue from the underlying connective tissue”?

Author’s response: Replaced as suggested.

Line 159, 161: maybe use the names laminin-332 for laminin-5 and laminin-511 for laminin-10?

Author’s response: Revised as suggested.

Line 185: Maybe mention the name batimastat for BB-94? Also note that this is not specific to MT1-MMP but a broad spectrum MMP inhibitor.

Author’s response: Revised as suggested.

Line 197: Furthermore, lumican is a competitive inhibitor of collagen cleavage by MT1-MMP (PMID: 25304424).

Author’s response: We appreciate the reviewer for telling us this interesting fact. We added the following sentence to section 2.1.5. “Interestingly, lumican was also identified as a competitive inhibitor of the MT1-MMP.”

Line 206: There are various collagen types. Not all of them are known to be substrates of MMP-2. Also, MMP-2 can cleave only monomeric collagen chains after at least partial unwinding of its triple helix.

Author’s response: Thanks for pointing this out. We added ‘type IV’.

Line 239: As described here, VEGFR-1 acts as a decoy receptor for VEGF-A; but it is also the exclusive receptor or other members of the VEGF family, such as VEGF-B and PlGF.

Author’s response: We added the following sentence as suggested. “It is also known that VEGFR-1 is a receptor of other members of the VEGF family, such as VEGF-B and PIGF.”

Line 262, Fig. 6: Why are small debris fragments shown only for ADAM-9 cleavage but not for VGFR-1 cleavage, and why are they red/blue colored? Also, in Figures A and B, why is the phosphorylated receptor shown again on a smaller scale at the far right?

Author’s response: We removed small debris fragments from the figure. And smaller scale ones were removed because they are basically the same but we wanted to show that there are multiple ones.

Line 320: Which types of cancer are meant here? Also, a reference would be desirable.

Author’s response: We specified the type of cancer, and the relative reference is added.

Line 343: Please specify what is meant with “active forms of proangiogenic factors”: ECM-tethered cytokines released by matrix degradation and/or ECM fragments acting as matrikines?

Author’s response: We appreciate the reviewer for this wise suggestion. We added TGF-β as an example of ECM-tethered cytokines.

Line 602: This citation is incomplete.

Author’s response: It has been completed now. Thanks!

English is fine. There are only a few minor grammar mistakes, e.g., lines 222, 317 f.

Author’s response: We thank the reviewer. Corrected.